# Unsupervised Learning in Drug Design from Self-Organization to Deep Chemistry

**DOI:** 10.3390/ijms23052797

**Published:** 2022-03-03

**Authors:** Jaroslaw Polanski

**Affiliations:** Institute of Chemistry, Faculty of Science and Technology, University of Silesia, Szkolna 9, 40-006 Katowice, Poland; polanski@us.edu.pl

**Keywords:** drug design, deep learning, deep chemistry, self-organizing maps, unsupervised learning, supervised learning, feature engineering, feature learning, molecular representation

## Abstract

The availability of computers has brought novel prospects in drug design. Neural networks (NN) were an early tool that cheminformatics tested for converting data into drugs. However, the initial interest faded for almost two decades. The recent success of Deep Learning (DL) has inspired a renaissance of neural networks for their potential application in deep chemistry. DL targets direct data analysis without any human intervention. Although back-propagation NN is the main algorithm in the DL that is currently being used, unsupervised learning can be even more efficient. We review self-organizing maps (SOM) in mapping molecular representations from the 1990s to the current deep chemistry. We discovered the enormous efficiency of SOM not only for features that could be expected by humans, but also for those that are not trivial to human chemists. We reviewed the DL projects in the current literature, especially unsupervised architectures. DL appears to be efficient in pattern recognition (Deep Face) or chess (Deep Blue). However, an efficient deep chemistry is still a matter for the future. This is because the availability of measured property data in chemistry is still limited.

## 1. Introduction

The availability of computers has brought novel prospects in drug design. The tautological term *rational drug design* (irrational design would be contrary to logic), coined for computer technologies, illustrates the high expectations in this area. However, after a few years of early fascination, in the late 1990s, medicinal chemists observed that the in silico methods did not live up to their promise. New ideas for increasing the efficiency of *computer-assisted drug discovery and development* were needed [1]. This reflection inspired the formation of cheminformatics. Because cheminformatics currently attempts to organize all of the research that connects chemistry and computer science, we often forget that drug design was its first task. 

Neural networks (NN) were an early tool that cheminformatics tested for converting data into drugs. Predicting properties and molecular mapping were among the applications. However, the initial interest faded for almost two decades. The confidence in NN waned. The methods seemed to be too obscure to support rational methods. It was only recently when there has been a renaissance of NNs. An NN is still a black box, but at the same time, it behaves like a magic tool. The success of deep learning (DL), an NN method that insists that machine learning can solve problems by learning from experience, is among the concepts that have caused this effect. Automated drug design is a novel paradigm and a priority [2]. DL performs direct data analysis without any human intervention. DL appears to be surprisingly efficient in pattern recognition (Deep Face), language translation, and chess: after a single win by Kasparov over Deep Blue in the late 1990s, no human has succeeded against a machine. 

Generally, current applications of unsupervised learning in DL are still rare. However, in drug design, unsupervised architectures can be surprisingly broadly observed, indicating the efficiency of the method and the fact that we need to process sizeable molecular data when measured properties are not available. This publication reviews recent applications of the DL algorithms for drug design, comparing them to the early unsupervised neural networks. In particular, in unsupervised learning applications for mapping molecular representations, we can still recognize the early neural network protoplasts. 

## 2. Artificial Intelligence, Machine or Deep Learning—Magic Tools or a Viral Buzz

Artificial intelligence (AI) is a popular term that appears relatively early, describing our potential for imitating natural human capabilities with computers [3]. The precise meaning of AI is vague. AI engages computer sciences and a variety of humanities, e.g., psychology and neurology. In the more narrow meaning, McCarthy defines AI as the science and engineering of making intelligent machines, especially intelligent computer programs [4]. 

Machine learning is a method of data processing by various in silico algorithms. This term refers to various methods, including decision trees, naive Bayes classifiers, random forest, support vector machine, hidden Markov models, and other data processing algorithms capable of handling big data. Machine learning can involve supervised, unsupervised, or reinforcement learning, depending upon the targeted outcome of data processing. In supervised systems, we attempt to predict the output values represented by the so-called training labels. We focus on searching for natural patterns and structures within the data with unsupervised methods. We do not use any training labels here. In turn, in reinforcement learning, machines can interact with the environment and get a reward for a proper action or behavior [5]. 

The term Deep Learning was coined by Rina Dechter in 1986 [6] and gained popularity with Igor Aizenberg, who searched for the ability to learn higher-level features from raw input data using multiple layer neural network architectures [7]. Geoffrey Hinton from the University of Toronto and Google provided recent inspiration in this field [8]. Autonomic behavior without any human intervention is among the desired functionalities. We usually associate DL with back-propagation, a specific NN architecture that is widely used in DL systems. Although back-propagation enables a variety of DL applications to be developed, DL can also involve other neural architectures, in particular deep, SOM architectures, e.g., with convolutional layers for clustering and visualizing image data [9]. An excellent introduction to DL methods and perspectives can be found in the Hinton interview [8]. Schneider indicates that we should not overestimate DL’s magic wands, which is a reincarnation of the early neural network methods developed in the 1990s [2]. DL algorithms process big data. In other words, DL guides us through big data and avoids routine chemistry. However, when discussing the perspectives of automated drug design and discovery, Schneider signifies the importance of DL methods by indicating their pattern recognition capabilities, especially when patterns escape the medicinal chemistry rationale [2]. In turn, Bajorath enlightened more critical issues, concluding that: *we are still far from ‘true’ AI in discovery settings where algorithms would make compound decisions beyond human reasoning* [10,11]. Often much simpler classifiers (logistic regression, decision lists) after preprocessing can give comparable results to more complex classifiers, deep neural networks, boosted decision trees, and random forests [12]. We should not increase model complexity if not needed. 

## 3. From Chemical Compounds to Drugs and Materials: Defining the Problem

Molecular design in drug and materials discovery can be defined in a mathematical form as mapping molecular properties to descriptors P → S. This procedure is known as a direct (Q)SAR problem and is only very rarely realized [1]. Practically, the majority of drug design methods rely on S → P mapping in which we form a model that is hopefully predictive enough to design novel compounds (having a certain calculable S) from a series of active drug or material candidates (having a certain calculable S and measured properties). The predicted P values for the calculated S can be proven after new compounds are synthesized (Figure 1). A variety of methods can use the single or the multiple chemotype and property domains. In particular, QSAR and m-QSAR usually model single chemotype domains and property domains. On the other hand, the diversity-oriented synthesis DOS (FOS: function-oriented synthesis; BIOS: biologically oriented synthesis) uses multiple domains. In turn, the chemotype domain definition is not crucial for OMICS projects (genomics, proteomics, lipidomics). While the early NN approaches usually processed single-chemotype domains, the current approaches are multi-chemotype projects. The next question refers to the molecular representation that is to be used in the calculation. Molecular representations are descriptors or properties [13]. Two available options are feature engineering and feature learning [14]. The term *feature* indicates that we are following the lexicon of informatics more than that of chemistry. The meaning of feature is somewhere between a (chemical) property or descriptor and a variable. When contrasting engineering vs. learning features, we focus on the autonomic capabilities of an algorithm. In *feature engineering*, we need human intervention to design variables that are then analyzed by algorithms. In turn, computers should be fully autonomous in feature learning, which means that the algorithm selects the features from among the raw data. In the chemical context, feature engineering asks how to construct a molecular representation. Which data should represent chemical compounds in a model? *Feature learning* is an algorithm capable of autonomous *feature engineering* by computer, thereby enabling the molecular representation that is suitable for the individual project to be determined.

Especially in the context of DL, the ability of *feature learning* is a critical issue because DL should be able to autonomously select features, i.e., molecular representations. Efficient *feature learning* still seems to be a matter for the future *deep chemistry*, while deep chess or deep face applications are now commonly available. Let us try to answer the question of what the reason for that is. More or less, DL processes big data. On the one hand, the ability to learn higher-level features is an advantage. On the other hand, we need big high-quality data to train a network. The human population is almost eight billion, and as many potential face data are readily available. By comparison, we only have ca. 2000 registered drugs (new molecular entities) and the registered chemical compounds count millions (more than 200,000,000 compounds). We have drug candidate databases that collect millions of chemical structures and their properties (ChEMBL, PubChem, ZINC), but not all of the data therein are measured properties. If we use databases in materials discovery, the data availability is even lower [15]. 

Schneider reported 70 million single SAR data points to illustrate how big the available data are [2]. Errors in the data are also a problem. For example, protein X-ray data needs the so-called data curation before use. In chess, a player’s errors will result in defeat, thus providing a clear signal to the deep algorithm. In turn, there is no straightforward relationship for the raw drug or materials data and project results, which are often uncertain. Chemistry and drug and materials discovery is a soft science. Therefore, the current practice still needs human *feature engineering*.

Chuang et al. [15] indicated the essential features of molecular representations necessary for medicinal chemistry. Molecular representations that are used for data processing should be (i) *expressive*, i.e., capable of coding an entire diversity of molecular data; (ii) *parsimonious*, simple but not too simple; (iii) *invariant*, should not change, for example, with a changing atom numbering pattern and (iv) *interpretable*; humans should be able to interpret the rules that are discovered by machine learning in order to easily find those that describe the data and not the artifacts or noises. Chuang et al. [15] also enumerated the human interpretable representations of molecules: (i) a bond-like notation with an atom as the vertex and bonds as the edges; (ii) 3D visual representations; (iii) multiple conformers (aligned poses); (iv) canonical SMILES and (v) computed molecular descriptors.

Grebner et al. evaluated the molecular representations available for virtual screening in drug design. How big are the representations that they form and how big is big enough? Virtual means that we screen billions of molecular representations. Comparing the novelty vs. accuracy vs. calculation speed for 2D, 3D and structure-based representations indicated that novelty and accuracy increases from 2D to SBDD while speed increases in the opposite direction. Because economy is essential in drug design, the authors compared the timings and estimated CPU/GPU costs for various representations (SBDD). For example, generating the conformer sets for 10^10^ molecules using the cloud-based workflow ORION technology is feasible within two to three days and can cost 20,000 USD, while a 3D high-quality comparison using the FastROCS method can cost as little as 100 USD per query [16]. 

## 4. Self-Organizing Mapping of Molecular Representations

Basically, neural networks (NN) are computer algorithms based on an alleged similarity to the human brain. A reader can find a brief but illustrative introduction to the chemical applications in the early references [17] or [18]. Figure 2 illustrates the differences in the supervised vs. unsupervised architectures. In both methods, we present molecular representations to the subsequent inputs and optimize the network to minimize the errors between the expected and actual output produced (supervised learning) or between the similarity of the signals and output. Supervised learning requires that the inputs involve labels, i.e., specific data for error optimization, while in unsupervised learning, the error is minimized by comparing the individual inputs. The details of the individual methods and examples of their applications can be found in many references, e.g., [17,18].

Intuitively, a molecular surface is an area that determines the drug-receptor interactions. Actually, the molecular surface is a representation that is of essential importance for drug design. In the early 1990s, Zupan and Gasteiger designed a scheme for mapping 3D molecular surfaces to a 2D representation. An application of the torus topology in this operation enabled the 3D topology to be fully preserved within a 2D map [17,18] Technically, the whole molecular surface can be observed within the map, the one that is normally seen from the observer’s point of view and the side of the molecule that is normally hidden from the observer. The maps were colored by electrostatic potential. Because the molecular surface and its electrostatic potential are closely associated with drug-receptor interactions, they mapped several molecules in an attempt to find the similarities between the drugs that stimulated the associated receptors [18,19].

An interesting feature of the SOM network is its ability to compare molecular surfaces [19,20]. In Figure 3, the surfaces of butane and propane are compared. Colors of the molecular fragments code the respective methyl (CH_3_) or methylene (CH_3_) formations. The answer to the question about the difference between butane and propane is an obvious chemical routine. Propane and butane are members of a homological series with a clear difference of single methylene (CH_2_). It is the same answer from chemical analysis; the formal chemical matter difference will amount to the weight of CH_2_. However, the answer from topology is not so obvious. If we superimpose the molecules without cutting them, then the difference is that the terminal methyl group of the larger butane will not find its counterpart in the propane molecule. However, this answer is not so clear because, in a butane vs. propane superimposition, the propane CH_3_ (four atoms) meets the butane CH_2_ (three atoms). Despite this uncertainty, the network identifies the disparity of the molecular pair as the lack of an uninterrupted surface correspondence (the upper part of Figure 3b). However, if the network parameters are changed, a pair of comparative maps can be observed (Figure 3b, bottom), which is a surprise for a chemist. The difference is now three distinct wholes on the surface. This picture needs careful analysis in order to understand the network signal. Accordingly, the hydrogen of the propane terminal CH_3_ group can take a *similar* position to the carbon atom of the terminal methyl of butane (Figure 3). This comparison can be identified as a *fuzzy topology*. Therefore, the SOM network not only indicates a feature that is trivial for humans or what they might expect (topology) but also a feature that one should also perceive but is overlooked, for example, due to a routine (fuzzy topology). Can this ability be used in drug design? Usually, the molecules are superimposed before the SOM comparison. However, in the comparative mapping of molecules discovered serendipitously in the Technical University of Munich in the early 1990s, a series of CBG steroid data was directly projected onto an SOM network that had been trained with the most active CBG analog. The data were used without any preprocessing or superimposition. The molecular surface was used directly as a result of the 3D simulator CORINA. The resulting series of comparative SOMs (Figure 4) are amazing. All of the low-activity compounds are highly white (full with empty neurons), while those with a high (H) or medium (M) activity are colored (black in the white and black representation). Interestingly, compound 21 was presented in the original paper with an error and provided a map full of whites, but after correction, the proper map was obtained, which may reveal an unbelievable competence of the simple SOM architecture for drug discovery. It should be remembered that the chemotypes of the series are not significantly diversified and that the rigid steroid structures form clear shape patterns. The 1990s were not a time when the chemical audience could accept that a comparison of molecules that were not superimposed could bring any informative results [21]. We published a study of the SOM patterns of fully superimposed structures [19] or the auto-correlation function for coding 3D CBG steroids along with the comparative SOM architecture [22]. Later, we developed the Comparative Molecular Surface Analysis coupling SOM with a PLS analysis, a method similar to the Comparative Molecular Field Analysis for modeling 3D QSARs [23,24,25]. CoMSA can be interpreted as being a fuzzy complementation of CoMFA.

Because molecular surfaces are generated from atomic 3D coordinates and atomic radiuses, the crude atomic representation could be used to feed the SOM network. A receptor-like neural network is an SOM network that has been fed with 3D atomic data [26] in which the NN is learning the data of the most active analog as the best-known template that resembles the receptor. Then, this network processes the 3D atomic coordinates of the other CBG series that have not been superimposed. Similar SOM structures will be developed for the atomic coordinates and the similarity of resulting maps will depend on the similarity to the atomic coordinates of the most active analog. As the atomic coordinates are processed, the molecular shape is a factor that limits the pattern of the map. Interestingly, when the atomic charges and not the shape are the limiting factors (testosterone binding globulin, TBG), the method fails and must be redesigned to simulate the so-called induced-fit drug-receptor interactions [26].

The atomic representations by the Cartesian coordinates are the essential pieces of information that were processed by the SOM architectures of the 1990s. In turn, contemporary deep chemistry most often uses SMILES, simple and computer-ready interpretable data. SMILES are, however, linear, while molecules are 3D objects. From this point of view, atomic coordinate data map the molecular shape landscape more naturally than SMILES. For a discussion on the use of SMILES in deep chemistry see reference [27]. 

4D QSAR is a method that uses multiple conformer ensembles for ligand-based molecular design [28]. For a recent review, compare the references [29,30]. Molecular dynamics are used to generate the so-called pose (multiple conformer-like) representations. The method uses voxels, i.e., small cubics, to define the spatial location of individual ligand atoms. Replacing the classical 4D QSAR voxels with the SOM representations (4D SOM-QSAR) improves the efficiency and stability of the method [31,32,33,34]. It also improves its predictive power.

Finally, an SOM network operates as a clustering tool, which forms a latent-like space and even in early applications could process large molecular representations of the size of 10^5^ [35]. This scheme has been popular in recent deep architectures. The early examples are mapping the 3D atomic data of HIV1 integrase inhibitors [35] or dopamine vs. benzodiazepine agonists [36]. For the topographic version, see reference [37]. In such an application, the network is fed with the molecular data for the whole library of molecules. An SOM network trained on a series of chemical compounds with a known functionality or activity level (training series) distributes them within the map. Then, the trained network used to cluster the designed molecules enables the activity for these novel analogs to be predicted depending on the similarity to the training library. This method, which is used for high throughput virtual screening, is both quick and relatively efficient. For example, for HIV1 integrase inhibitors the network was used to project in latent space 26,784 virtual compounds [35]. A variety of SOM modifications known as topographic mappings were published in the late 1990s by the group of Bishop [38]. The introduction to this method in the context of drug design and a comprehensive review can be found in the reference [37]. Recently, Qian et al. discussed the perspectives for using SOM in materials design mainly as a clustering tool [39]. 

## 5. Deep Learning for Processing Molecular Data in Drug Design

The potential profits of DL in rational drug discovery were recently reviewed in the references [40,41,42,43]. Multilayer structures enable the extraction of cascade features that work with nonlinear functions [2]. The main DL algorithm is a multilayer back-propagation that has been optimized for various tasks. Hinton explains the efficiency of this method by the fact that much effort has been expended to optimize it [8]. He also signifies that other architectures, particularly the unsupervised ones, can appear to be even more efficient. Table 1 presents individual examples of DL representations in drug design and indicates the supervised and unsupervised schemes. We can observe that unsupervised schemes are used surprisingly broadly. Probably, this indicates not only the efficiency of the method but also the fact that we need to process a sizeable molecular data share that is virtually generated, i.e., the measured properties do not label this portion of the data. The DL lexicon uses the term generative models (generative chemistry) for unsupervised algorithms to stress the difference between the classical design based on the local domain molecular exploration vs. the DL systematic continuous screening. Born and Manica [42] predicted that multimodal deep learning chemistry using disparate sources to generate molecules would be the next challenge in DL in the near future. The Variational Autoencoder (VAE) method [44] was developed as an algorithm to learn continuous molecular representations. This method was used by Gomez-Bombarelli et al. for an automatic chemical design using a data-driven continuous representation of molecules. In the critical operation of the latent space formation, the architecture analyzes the similarity of the SMILES codes of the candidate and the known inhibitor structures. A deep neural network involves three coupled functions: an encoder, a decoder and a predictor (Figure 5). The encoder converts the SMILES data into a continuous-like molecular representation, forming the latent molecular space in unsupervised learning. The distance in the space from the known highly active molecules defines the drug-likeness potential of the candidate structure. Such a latent representation enables the automatic generation of novel structures by perturbing or interpolating between the input chemical structures. Because SMILES codes represent molecules, this space can easily be decoded back to discrete molecular representations. On the other hand, the supervised perceptron algorithm predicts the biological properties from the latent space representation [45].

Self-organizing maps (SOMs) are used to quickly identify the potent DDR1 kinase inhibitors in the DL method. The SOM-based reward function scores the compound novelty based on the known DDR1 kinase inhibitors and patented structure data. The entire study, which involved a final synthesis and testing, was a 46-day-long project [46]. The authors used six data sets (1) a large ZINC-extracted data set, (2) known DDR1 kinase inhibitors, (3) common kinase inhibitors, (4) active compounds vs. non-kinase targets, (5) patented DDR1 kinase candidate structures and (6) 3D structures for the DDR1 inhibitors. Data processing involved operations such as (1) general and specific kinase self-organizing mapping (SOM), (2) modeling the pharmacophore by the crystal structures of the compounds in a complex with DDR1 or Sammon mapping. The study involves 40 structures (randomly selected) that covered the resulting chemical space and the distribution of the RMSD values. A chemical synthesis proved the calculation results. 

**Table 1 ijms-23-02797-t001:** Recent DL applications in drug design.

Problem	Data/Learning Type	Reference
DNA subregion binding	In vitro HTS/convolutional neural networks	[47]
Protein function	3D electron density/convolutional filters	[48]
Genomics	Gene expression contrastive divergence (unsupervised) [49]; back-propagation (supervised) [50]; multilayer perceptron [51] supervised	[49,50,51]
Pharmacodynamics (DeepDTI)	Drug-protein interaction/unsupervised/then supervised [52]; supervised [53]	[52,53]
DeepAffinity	Compound-protein affinity/supervised	[54]
DeepTox toxicity	Toxic data/multi-task networks (supervised)	[55]
Drug IC50	Mol. descriptors/supervised	[56]
VAE chemical properties	SMILES; molecular graphs/unsupervised	[45,57,58,59,60]
VAE/GENTRL DDR1 small molecule design	SMILES; Kohonen-SOM based reward function/semi-supervised	[46]
VAE/Graph encoders	Molecular graphs/unsupervised	[61,62,63]
Protein-ligand pair	SMILES; voxels/unsupervised	[64,65]
CMap/gen perturbagens	Gen-expression profiles/unsupervised	[66]
Scaffold generation	molecular graphs; physicochemical properties; fragments/unsupervised	[67,68,69,70,71,72,73,74,75]

Blaschke et al. [76] indicates that although generative modelling was applied in de novo design of novel active ligands [46] chemical diversity of the compounds often imitates to closely previous chemotypes [77,78,79].

Property prediction and molecular modeling are autonomous areas related to drug design. The prospects of applying deep learning in property prediction were reviewed recently by Walters and Barzilay [80]. The main conclusion is that training data is critical in generating any machine learning model. Although we have large property databases (PubChem or ChEMBL), the quality of these data can sometimes be questionable. In turn, pharmaceutical company data are structured inconsistently and not shared eagerly and therefore are hard to use in predictive property modeling. The early neural network models used fingerprints or other molecular descriptors as molecular representations. Then we should weigh the contribution of the individual features within the model, or in other words, we use feature engineering schemes. Current representations targeted at property prediction can learn features directly from the data, mapping SMILES or graphs into dense continuous vectors. Generative modeling allows de novo molecular design. The encoded SMILES or molecular graphs are mapped into the latent molecular space, which, unlike typical discrete representations, is designed to be smooth. Practically, the smoothness of such latent maps could, however, be questioned. [80]. This fact also indicates the limitation of SMILES which are linear, while molecules are 3D objects. Moreover, the SMILES of very similar objects can be completely different. In turn, the space 3D coordinates seem to be more natural. Such representations are typically used for noting X-ray structures. The 3D coordinates naturally represent molecular space, although the multi-conformation can be a problem. However, we should remember that deep learning operates by comparing similarities so that similar structures will generate similar conformations. 

Molecular modeling is another exciting area for applying deep learning; for a recent review, see reference [81]. In molecular modeling, molecules (small molecules or macromolecules) are handled as geometric representations in 3D Euclidean space. Molecular representations used to simulate deep learning models are SMILES or sparse molecular graphs or amino acid sequence data. Deep learning-based molecular modeling could improve navigating chemical space, changing data mining in cheminformatics. For a detailed discussion of the problems, the reader should compare the review [81].

## 6. Feature Engineering vs. Feature Learning—A Lesson from Deep Retrosynthetic Approaches

Computer-assisted synthesis design (CASD) is an in silico application that has recently significantly profited from NN. CASD explores the potential ways to obtain a specific molecule by searching and probing a synthesis tree among potential reactions and reagents. *Retrosynthesis* is a method designed by Corey to solve this problem [82]. Corey also programmed the first software (LHASA, Harvard, Cambridge, MA, USA) to get computer assistance in the field. However, until recently, computers defeated the competition. Currently, there is tremendous interest in applying NN architectures in CASD [83,84]. DL is more and more competent here; however, humans still appear to be better at finding the critical disconnections within complex natural product molecules and human-machine cooperation wins the competition [85,86,87,88,89]. In conclusion, we still need feature engineering. However, neural networks have to support humans to succeed additionally. 

## 7. Conclusions

The development of efficient DL methods can be observed in deep face (face recognition) and deep blue (chess playing) recently. These developments inspired the rebirth of using neural networks in drug design. The data in drug design are getting bigger and bigger; the 70 million SAR data points can illustrate data availability here [2]. The DL algorithms fit big data processing well. The most crucial advantage of DL is its ability to operate autonomously, which is a priority when automated drug design is the novel paradigm of the targeted future of medicinal chemistry. The so-called feature learning, i.e., the ability for autonomous feature selection, is a central quality of DL. Computer-aided synthesis design (CASD) is an example of enormous developments in recent years. The lesson from CASD is that a full feature learning mode is still a matter for the future. The most efficient methods still need human engineering features. 

We should not overestimate DL’s *magic wands*, reincarnating the early neural network methods developed in the 1990s. Although DL algorithms currently use the supervised mode, the hope is that unsupervised algorithms could be even more efficient. We can observe that the unsupervised schemes are already used in drug design surprisingly broad. Probably, this indicates not only the efficiency of the method but also the fact that we need to process a sizeable molecular data share that is virtually generated, i.e., the measured properties do not label this portion of the data. This publication reviewed the current DL applications and compared them to the early unsupervised. 

## Figures and Tables

**Figure 1 ijms-23-02797-f001:**
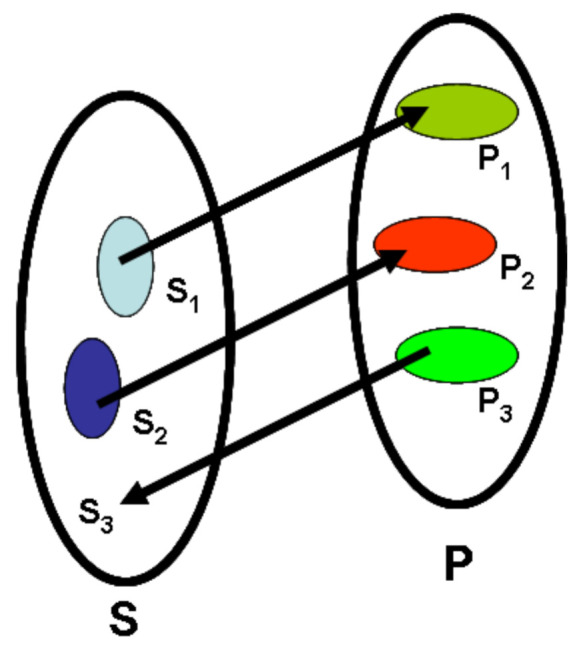
The direct drug design problem can be defined as mapping property to structure (P → S)_3_. Mainly, it is realized in the indirect mode by structure to property mapping (S → P). Individual methods allow to include various domains (S → P)_1_ or (S → P)_2_. Domain diversity is indicated schematically by colors.

**Figure 2 ijms-23-02797-f002:**
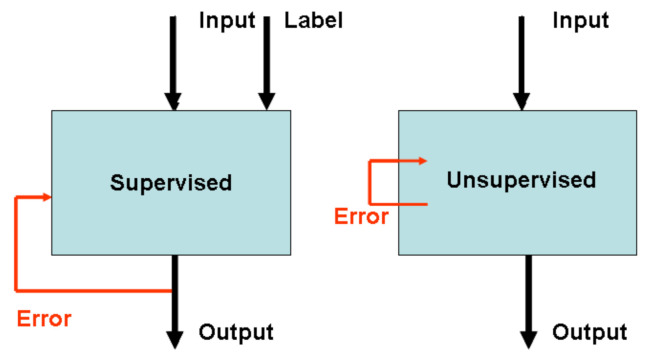
Supervised learning vs. unsupervised learning architectures. Both modes demand optimization; however, while in supervised learning, we need a label within the inputs which we use to estimate the error between the label and the output value, in unsupervised learning, the error is minimized by comparing the unlabeled inputs.

**Figure 3 ijms-23-02797-f003:**
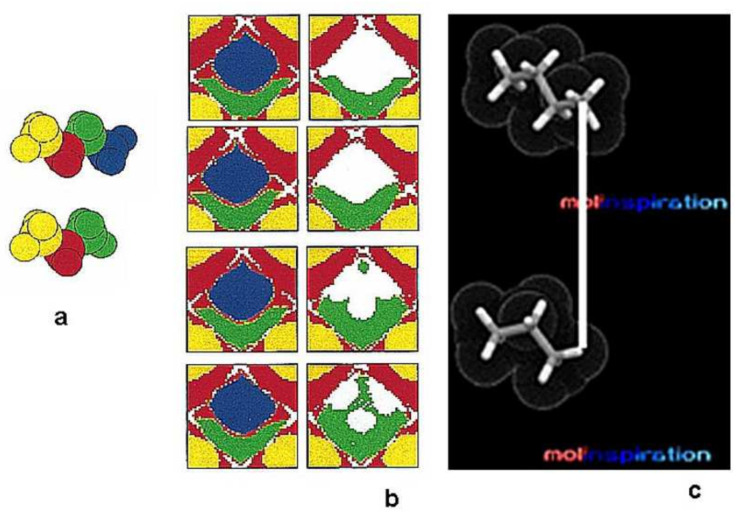
The propane vs. butane colored by methyl (yellow or blue in butane, yellow or green in propane) and methylene (red or green in butane, red in propane) fragments (**a**) provides a series of two types of CoMSA (SOM) projections (**b**), depending upon the SOM network regulation. Two types of patterns (**b**) can be explained by fuzzy topology (**c**). Details in text.

**Figure 4 ijms-23-02797-f004:**
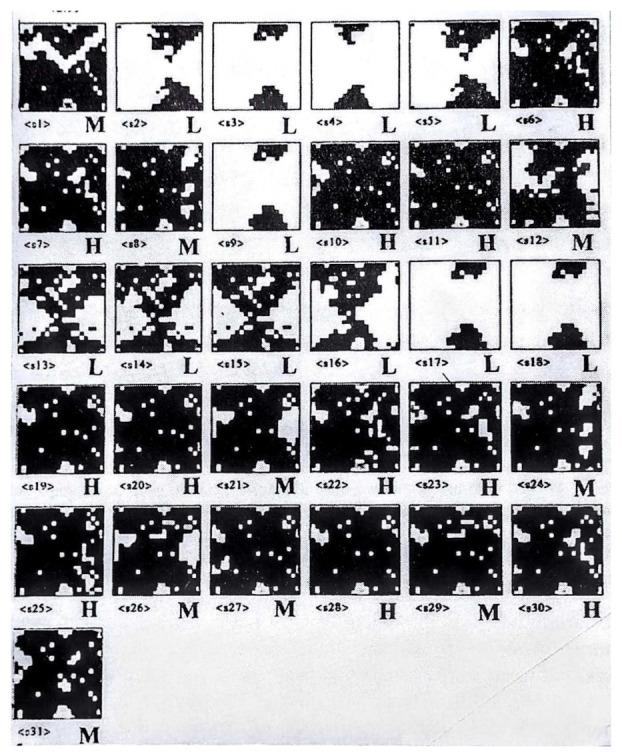
A series of CBG steroid surface data projected by CoMSA (SOM) without superimposition [21]. Without a single misinterpretation, H (high) and M (medium) activity compounds can be differentiated from the L (low) activity compounds. Details in text. Copyright © 1996 Polish Chemical Society.

**Figure 5 ijms-23-02797-f005:**
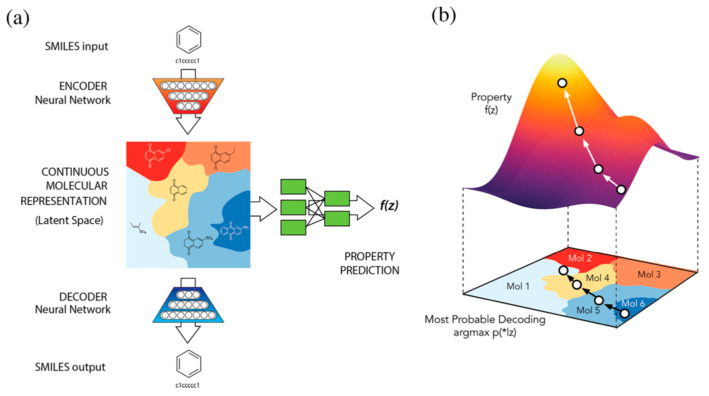
Automatic chemical design using a data-driven continuous representation of molecules. In the critical operation of the latent space formation, the architecture analyzes the similarity of the SMILES codes of the candidate and the known inhibitor structures. A deep neural network involves three coupled functions: an encoder, a decoder (**a**) and a predictor (**b**) [45]. Copyright © 2018 American Chemical Society.

## Data Availability

Not applicable.

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
