# Peer review of "Unsupervised Learning in Drug Design from Self-Organization to Deep Chemistry"

_ijms, 2022, doi:10.3390/ijms23052797_

Round 1
Reviewer 1 Report
The author has reported the DL projects in the current literature, especially the unsupervised architectures. DL appears to be efficient in pattern recognition (Deep Face) or when playing chess (Deep Blue). However, efficient deep chemistry is still a matter of the future. This is because the availability of measured property data in chemistry is still limited. In my opinion, the paper needs major revision.
- The abstract should be in one flow, no need to mention, Abstract: Background: Methods: Results: Conclusions separately.
- The introduction is quite premature and shallow. Ideally, readers expect to have a very brief account of the aims, methods, key findings, and conclusions of a study from an abstract with a couple of sentences from each part.
- The author should mention the Machine Learning Classifiers in detail e.g. KNN, Decision Tree (DT), Random Forest (RF), Naïve Bayes, in the main text.
- Also mention the Descriptor and Fingerprint Calculation should explore detail in the main text.
- Page no 7, line no 265, Deep learning for processing molecular data in drug design……..author should explain in detail about Deep learning for processing molecular modeling.
- English editing of language and style required to improve.
Author Response
1. The abstract should be in one flow, no need to mention, Abstract: Background: Methods: Results: Conclusions separately.
Thank you very much for this remark. We corrected the abstract accordingly.
The availability of computers has brought novel prospects in drug design. Neural networks (NN) were an early tool that cheminformatics tested for converting data into drugs. However, the initial interest faded for almost two decades. The recent success of Deep Learning (DL) has inspired a renaissance of neural networks for their potential application in deep chemistry. DL targets direct data analysis without any human intervention. Although back-propagation NN is the main algorithm in the DL that is currently being used, unsupervised learning can be even more efficient. We review self-organizing maps (SOM) in mapping molecular representations from the 1990s to the current deep chemistry. We can discover the enormous efficiency of SOM, not only for features that could be expected by humans but also those that are not trivial to human chemists. We reviewed the DL projects in the current literature, especially the unsupervised architectures. DL appears to be efficient in pattern recognition (Deep Face) or chess (Deep Blue). However, the efficient deep chemistry is still a matter of the future. This is because the availability of measured property data in chemistry is still limited.
2. The introduction is quite premature and shallow. Ideally, readers expect to have a very brief account of the aims, methods, key findings, and conclusions of a study from an abstract with a couple of sentences from each part.
Thank you very much for this remark. We reedited this part removing text to a novel chapter.
2. Artificial intelligence, machine or deep learning – magic tools or a viral buzz
Artificial intelligence (AI) is a popular term that appears relatively early, describing our potential of imitating natural human capabilities by computers [3]. The precise meaning of AI is vague. AI engages computer sciences and a variety of humanities, e.g., psychology and neurology. In the more narrow meaning, McCarthy defines AI as the science and engineering of making intelligent machines, especially intelligent computer programs [4].
Machine learning is a method of data processing by various in silico algorithms. This term refers to various methods, including decision trees, naive Bayes classifiers, random forest, support vector machine, hidden Markov models, or other data processing algorithms capable of handling big data. Machine learning can involve supervised, unsupervised or reinforcement learning, depending upon the targeted outcome of data processing. In supervised systems, we attempt to predict the output values represented by the so-called training labels. We focus on searching for the natural patterns and structures within the data in unsupervised methods. We do not use any training labels here. In turn, in reinforcement learning, machines can interact with the environment and get a reward for a proper action or behavior [5].
The term Deep Learning was coined by Rina Dechter in 1986 [6] and gained popularity with Igor Aizenberg who searched for the ability to learn higher-level features from raw input data using multiple layer neural network architectures [7]. Geoffrey Hinton from the University of Toronto and Google provided recent inspiration in this field [8]. Autonomic behavior without any human intervention is among the desired functionalities. We usually associate DL with back-propagation, a specific NN architecture that is widely used in DL systems. Although back-propagation enables a variety of DL applications to be developed, DL can also involve other neural architectures, in particular deep, SOM architectures, e.g., with convolutional layers for clustering and visualizing image data [9]. An excellent introduction to the DL methods and perspectives can be found in the Hinton interview [8]. Schneider indicates that we should not overestimate DL's magic wands, which is a reincarnation of the early neural network methods developed in the 1990s [2]. DL algorithms process big data. In other words, DL guides us through big data and avoids routine chemistry. However, when discussing the perspectives of automated drug design and discovery, Schneider signifies the importance of DL methods by indicating their pattern recognition capabilities, especially when patterns escape the medicinal chemistry rationale [2]. In turn, Bajorath enlightened more critical issues, concluding that: we are still far from ‘true’ AI in discovery settings where algorithms would make compound decisions beyond human reasoning [10, 11]. Often much simpler classifiers (logistic regression, decision lists) after preprocessing can give comparable results than more complex classifiers, deep neural networks, boosted decision trees, random forests [12]. We should not increase model complexity if not needed.
3. The author should mention the Machine Learning Classifiers in detail e.g. KNN, Decision Tree (DT), Random Forest (RF), Naïve Bayes, in the main text.
This was mentioned as required:
Machine learning is a method of data processing by various in silico algorithms. This term refers to various methods, including decision trees, naive Bayes classifiers, random forest, support vector machine, hidden Markov models, or other data processing algorithms capable of handling big data. Machine learning can involve supervised, unsupervised or reinforcement learning, depending upon the targeted outcome of data processing. In supervised systems, we attempt to predict the output values represented by the so-called training labels. We focus on searching for the natural patterns and structures within the data in unsupervised methods. We do not use any training labels here. In turn, in reinforcement learning, machines can interact with the environment and get a reward for a proper action or behavior [5].
Also mention the Descriptor and Fingerprint Calculation should explore detail in the main text.
This was mentioned as a part of the novel property prediction paragraph, as required:
Property prediction and molecular modeling are autonomous areas related to drug design. The prospects of applying deep learning in property prediction were reviewed recently by Walters and Barzilay [80]. The main conclusion is that training data is critical in generating any machine learning model. Although we have large property databases (PubChem or ChEMBL), the quality of these data can sometimes be questionable. In turn, pharmaceutical company data are structured inconsistently and not shared eagerly and therefore are hard to use in predictive property modeling. The early neural network models used fingerprints or other molecular descriptors as molecular representations. Then we should weigh the contribution of the individual features within the model, or in other words, we use feature engineering schemes. Current representations targeted at property prediction can learn features directly from the data, mapping SMILES or graphs into dense continuous vectors. Generative modeling allows de novo molecular design. The encoded SMILES or molecular graphs are mapped into the latent molecular space, which, unlike typical discrete representations, are designed to be smooth. Practically, the smoothness of such latent maps could, however, be questioned. [80].
4. Page no 7, line no 265, Deep learning for processing molecular data in drug design……..author should explain in detail about Deep learning for processing molecular modeling.
This was discussed accordingly:
Molecular modeling is another exciting area for applying deep learning; for a recent review, see reference [81]. In molecular modeling, molecules (small molecules or macromolecules) are handled as geometric representations in 3D Euclidean space. Molecular representations used to simulate deep learning models are SMILES or sparse molecular graphs or amino acid sequence data. Deep learning-based molecular modeling could improve navigating chemical space, changing data mining in cheminformatics. For a detailed discussion of the problems, the reader should compare the review [81].
English editing of language and style required to improve.
English language and style was improved.

Reviewer 2 Report
This review is of interest. I have minor suggestions below
It is correct to say that 3D objects could be more valuable than Smiles but the problem is which 3D structure to use ? Many molecules are flexible and some change conformation upon binding to a receptor, thus many errors can occur with 3D and at the end, what is best, error due to limitations in the SMILES format or errors in 3D. Explanations are missing.
Of course ones can use multiple conformers, but how to deal with a collection of billions of molecules then ?
It might be needed to give an example of SOM on for instance fingerprints and show the output and how it can help drug design.
The importance of the study in ref 40 has been largely discussed by many actors, like in the forum of Derek Lowe (https://www.science.org/blogs/pipeline), by the group of Andreas Bender (on a blog or in a publication..), in the blog of Pat Walters… The overall message from these people is that in ref 40, they identified a compound that is highly similar (one aromatic ring difference) to a DDR1 inhibitor present I think it was ChEMBL (maybe the molecule is also in clinical trial..). As such, a simple fingerprint similarity search shows that their no innovation. Maybe it was Pat Walters who mentioned that innovation should be considered the same way, for computer design and for molecules developed by medicinal chemists, suggesting that most deep learning study in the field of drug design are overselling the results and make massive buzz … The improvements in Deep learning for instance for ADMET predictions or protein-ligand interactions are essentially overfitting or related but the results do not generalize because of numerous reasons including the lack of labelled data and experimental errors. Several articles for instance from the Bajorath group comment this point. Maybe some additional paragraphs about these views, largely accepted in the community but by AI startups that are making millions with intoxicating news like from start to the clinic in 3 months, 1 month etc… would be valuable for the readers of this review.
Author Response
It is correct to say that 3D objects could be more valuable than Smiles but the problem is which 3D structure to use ? Many molecules are flexible and some change conformation upon binding to a receptor, thus many errors can occur with 3D and at the end, what is best, error due to limitations in the SMILES format or errors in 3D. Explanations are missing.
Of course ones can use multiple conformers, but how to deal with a collection of billions of molecules then ?
Thank you for this remark. Accordingly, we comment the problem.
Practically, the smoothness of such latent maps could, however, be questioned. [80]. This fact also indicates the limitation of SMILES which are linear, while molecules are 3D objects. Moreover, the SMILES of very similar objects can be completely different. In turn, the space 3D coordinates seem to be more natural. Such representations are typically used for noting X-ray structures. The 3D coordinates naturally represent molecular space, although the multi-conformation can be a problem. However, we should remember that deep learning operates by comparing similarities so that similar structures will generate similar conformations.
It might be needed to give an example of SOM on for instance fingerprints and show the output and how it can help drug design.
Thank you for this remark. Accordingly, we comment the problem, referring the reader to the additional reference. Additionally, we discussed property prediction problem, i.e., the application of fingerprint to property modeling.
The early neural network models used fingerprints or other molecular descriptors as molecular representations. Then we should weigh the contribution of the individual features within the model, or in other words, we use feature engineering schemes. Current representations targeted at property prediction can learn features directly from the data, mapping SMILES or graphs into dense continuous vectors. Generative modeling allows de novo molecular design. The encoded SMILES or molecular graphs are mapped into the latent molecular space, which, unlike typical discrete representations, are designed to be smooth. Practically, the smoothness of such latent maps could, however, be questioned. [80].
Property prediction and molecular modeling are autonomous areas related to drug design. The prospects of applying deep learning in property prediction were reviewed recently by Walters and Barzilay [80]. The main conclusion is that training data is critical in generating any machine learning model. Although we have large property databases (PubChem or ChEMBL), the quality of these data can sometimes be questionable. In turn, pharmaceutical company data are structured inconsistently and not shared eagerly and therefore are hard to use in predictive property modeling. The early neural network models used fingerprints or other molecular descriptors as molecular representations. Then we should weigh the contribution of the individual features within the model, or in other words, we use feature engineering schemes. Current representations targeted at property prediction can learn features directly from the data, mapping SMILES or graphs into dense continuous vectors. Generative modeling allows de novo molecular design. The encoded SMILES or molecular graphs are mapped into the latent molecular space, which, unlike typical discrete representations, are designed to be smooth. Practically, the smoothness of such latent maps could, however, be questioned. [80].
The importance of the study in ref 40 has been largely discussed by many actors, like in the forum of Derek Lowe (https://www.science.org/blogs/pipeline), by the group of Andreas Bender (on a blog or in a publication..), in the blog of Pat Walters… The overall message from these people is that in ref 40, they identified a compound that is highly similar (one aromatic ring difference) to a DDR1 inhibitor present I think it was ChEMBL (maybe the molecule is also in clinical trial..). As such, a simple fingerprint similarity search shows that their no innovation. Maybe it was Pat Walters who mentioned that innovation should be considered the same way, for computer design and for molecules developed by medicinal chemists, suggesting that most deep learning study in the field of drug design are overselling the results and make massive buzz … The improvements in Deep learning for instance for ADMET predictions or protein-ligand interactions are essentially overfitting or related but the results do not generalize because of numerous reasons including the lack of labelled data and experimental errors. Several articles for instance from the Bajorath group comment this point. Maybe some additional paragraphs about these views, largely accepted in the community but by AI startups that are making millions with intoxicating news like from start to the clinic in 3 months, 1 month etc… would be valuable for the readers of this review.
Thank you very much for this remark. We completely agree with this remark. For example, the whole information of the text is that we often overlook the similarity of novel methods to early nn methods. Accordingly, we added a new paragraph:
-
Artificial intelligence, machine or deep learning – magic tools or a viral buzz
[…..] Schneider signifies the importance of DL methods by indicating their pattern recognition capabilities, especially when patterns escape the medicinal chemistry rationale [2]. In turn, Bajorath enlightened more critical issues, concluding that: we are still far from ‘true’ AI in discovery settings where algorithms would make compound decisions beyond human reasoning [10, 11]. Often much simpler classifiers (logistic regression, decision lists) after preprocessing can give comparable results than more complex classifiers, deep neural networks, boosted decision trees, random forests [12]. We should not increase model complexity if not needed.
and further remark on this topic:
Blaschke et al. [76] indicates that although generative modelling was applied in de novo design of novel active ligands [46] chemical diversity of the compounds often imitates to closely previous chemotypes [77-79].

Round 2
Reviewer 1 Report
Author have mentioned all the answer properly.